# Accelerating Chloroplast Engineering: A New System for Rapid Generation of Marker-Free Transplastomic Lines of *Chlamydomonas reinhardtii*

**DOI:** 10.3390/microorganisms11081967

**Published:** 2023-07-31

**Authors:** Henry N. Taunt, Harry O. Jackson, Ísarr N. Gunnarsson, Rabbia Pervaiz, Saul Purton

**Affiliations:** 1Department of Structural and Molecular Biology, University College London, Gower Street, London WC1E 6BT, UK; 2Centre of Excellence in Molecular Biology, University of the Punjab, Lahore 53700, Pakistan

**Keywords:** *Chlamydomonas reinhardtii*, chloroplast, photosynthetic restoration, synthetic biology, transformation

## Abstract

‘Marker-free’ strategies for creating transgenic microorganisms avoid the issue of potential transmission of antibiotic resistance genes to other microorganisms. An already-established strategy for engineering the chloroplast genome (=plastome) of the green microalga *Chlamydomonas reinhardtii* involves the restoration of photosynthetic function using a recipient strain carrying a plastome mutation in a key photosynthesis gene. Selection for transformant colonies is carried out on minimal media, such that only those cells in which the mutated gene has been replaced with a wild-type copy carried on the transgenic DNA are capable of phototrophic growth. However, this approach can suffer from issues of efficiency due to the slow growth of *C. reinhardtii* on minimal media and the slow die-back of the untransformed lawn of cells when using mutant strains with a limited photosensitivity phenotype. Furthermore, such phototrophic rescue has tended to rely on existing mutants that are not necessarily ideal for transformation and targeted transgene insertion: Mutants carrying point mutations can easily revert, and those with deletions that do not extend to the intended transgene insertion site can give rise to a sub-population of rescued lines that lack the transgene. In order to improve and accelerate the transformation pipeline for *C. reinhardtii*, we have created a novel recipient line, HNT6, carrying an engineered deletion in exon 3 of *psaA*, which encodes one of the core subunits of photosystem I (PSI). Such PSI mutants are highly light-sensitive allowing faster recovery of transformant colonies by selecting for light-tolerance on acetate-containing media, rather than phototrophic growth on minimal media. The deletion extends to a site upstream of *psaA-3* that serves as a neutral locus for transgene insertion, thereby ensuring that all of the recovered colonies are transformants containing the transgene. We demonstrate the application of HNT6 using a luciferase reporter.

## 1. Introduction

DNA transformation of the *Chlamydomonas reinhardtii* chloroplast has been possible since the late 1980s [1] and since then has been used to probe many aspects of chloroplast physiology including the assembly and function of the photosynthetic apparatus [2], comparative analysis of photosystem subunits [3], and chloroplast gene expression and regulation [4]. The algal chloroplast also represents a potential platform for biotechnological exploitation given its enclosed sub-cellular architecture, the high ploidy and expression level of its genes, the ability to precisely target transgenes into its small genome (=plastome) via homologous recombination, and the lack of silencing mechanisms [5]. Furthermore, biotechnology using phototrophic microorganisms such as *C. reinhardtii* offers the potential of a sustainable green platform requiring only light, CO_2_, and basic nutrients for cultivation [6]. 

One key consideration of any microbial cell platform used in biotechnology is the presence in the genome of transgenes encoding antibiotic resistance that are introduced as selectable markers during transgenesis, and could spread to other microorganisms via horizontal gene transfer [7]. Given the chloroplast’s evolutionary history as an endosymbiont of an early cyanobacterial cell, such antibiotic-based selectable markers designed for high-level expression in the chloroplast are often functional in bacteria [8]. When combined with the high ploidy of plastomes where as many as a hundred copies can be present in a single chloroplast [9], the spread of antibiotic resistance cassettes to the environment can be considered a very real danger [10]. As in other biotechnology platforms, such a risk can be mitigated by using a ‘rescue of loss-of-function’ strategy in which an auxotrophic mutant with a specific growth requirement is used as the host, and restoration of the wild-type phenotype is used as the selection. One example is the uracil auxotrophy system commonly used in *Saccharomyces cerevisiae* transformation [11]. In the case of *C. reinhardtii*, the cell’s ability to switch to heterotrophic growth in the dark when supplied with an exogenous carbon source such as acetate allows for the isolation of photosynthetic mutants, and hence the use of the wild-type allele of the affected endogenous gene as a selection marker for restoration of photosynthesis [12].

Several groups have taken advantage of the mutant collection available for *C. reinhardtii* to develop systems for transgene delivery into the plastome that avoid the use of antibiotic-based selection. These systems are based on the rescue of a photosynthetic mutant carrying a lesion in a chloroplast gene encoding a core component of one of the photosynthetic complexes. Examples of such lesions include deletions of *atpB*, *psbA*, or *tscA*, and point mutations in *rbcL* or *psaB* [8]. However, these strains were generated using traditional mutagenesis screens many decades ago and the nature of the lesions is not necessarily suited for chloroplast genetic engineering. Several issues, therefore, arise when using such strains as transformation recipients, in addition to uncertainties about the ‘vigor’ of such strains given the likelihood of additional mutations within the genome and genetic drift of other photosynthetic genes following many years of maintenance on acetate-containing media. Mutants carrying point mutations carry the risk of reversion or suppression during transformation selection, giving rise to a background of false-positive colonies. Furthermore, a point mutation or small deletion can be repaired by transformation (through homologous recombination on either side of the lesion) without integration of the linked transgene [8]. This is of particular concern where there is a negative selection pressure associated with the transgene that is being introduced, such as toxicity or metabolic burden. In contrast, large deletions such as those spanning *atpB* or *psbA* in mutant strains Fud50 and Fud7, respectively, avoid these problems. However, restoration requires that the entire deleted DNA plus flanking regions are included in the transformation vector to allow recombination on either side of the deletion [13]. This can result in a large size for the final plasmid, especially if multiple transgenes are to be introduced, and can make the plasmid construction and amplification in *E. coli* problematic. Finally, a neutral locus for transgene insertion is required within the deletion to insure the co-integration of both the photosynthetic gene and the transgene in all transformant lines [8]. 

Wannathong and colleagues [14] addressed these issues by creating a novel photosynthetic mutant, TN72, in which 359 nucleotides spanning most of the coding sequence and 3′ untranslated region (UTR) of the essential photosystem II gene *psbH* were replaced with the *aadA* cassette conferring spectinomycin resistance. This gave rise to a stable ΔPSII strain suitable for transformation whereby the mutant phenotype could be rescued using plasmids carrying a WT copy of the small *psbH* gene. Furthermore, the deleted plastome region was designed to extend as far as a well-established neutral locus for transgene insertion, ensuring that *psbH* restoration was always accompanied by transgene insertion. TN72 has been used by several groups to create marker-free transgenic lines producing recombinant therapeutics [14,15,16,17,18,19]. However, TN72 has several drawbacks as a platform strain. Firstly, PSII mutants of *C. reinhardtii* are much less sensitive to high light compared to mutants lacking PSI, the cytochrome *b*_6_*f* complex, or the ATP synthase [20]. This is because PSII is the major instigator of photooxidative damage under high-light conditions when the electron flow through the photosynthetic apparatus is blocked by mutation or inhibitors [21]. Consequently, selection for the rescue of the *psbH* mutant relies on plating transformed cells on a minimal medium lacking acetate and waiting for phototrophic colonies to appear against a background lawn of untransformed cells that slowly die back due to acetate starvation. As phototrophic growth of *C. reinhardtii* is much slower than mixotrophic growth (i.e., growth on acetate in the light), colonies can take up to six weeks to appear. The subsequent restreaking of primary transformants to obtain homoplasmic lines must also be performed on minimal medium, further slowing the process. Secondly, TN72 was generated using the cell wall-deficient strain *cw15*. Whilst this allows DNA delivery into the chloroplast using a simple and cheap vortexing method [22] rather than microparticle bombardment using an expensive gene gun, the vortexing method is much less efficient, typically resulting in only a few transformant colonies per plate compared to over a hundred colonies following bombardment [13]. Multiple plates are therefore required to recover sufficient independent transformant lines for analysis. Furthermore, the cell wall-deficient phenotype of TN72 makes it sensitive to shear forces and complicates the harvesting of intact biomass from large-scale cultures of transgenic lines [23].

To remedy these shortcomings, we have built a novel system based on the same design concept, but using a wildtype (i.e., walled) strain as the parental line, and targeting an essential PSI gene, *psaA-3*, rather than a PSII gene. Such ΔPSI mutants are highly light-sensitive [24]. Consequently, the rescue of the mutant can be based on tolerance to high light on acetate-containing medium, allowing colonies to form in approximately six days following microparticle bombardment. Compared to previous systems, this represents a significant saving in both time and resources.

## 2. Materials and Methods

### 2.1. Cultivation of Algal Lines

*C. reinhardtii* lines used in this study were constructed from the wild-type strain 21gr mt+ (=CC-1690), supplied by the Chlamydomonas Resource Center (University of Minnesota, Minneapolis, MN, USA), and were grown on Tris Acetate Phosphate (TAP) or High Salt Minimal (HSM) media as appropriate [25]. Stock strains were maintained on solid media supplemented with 1.5% agar and were cultivated at 25 °C under 24 h ~50 μE s^−1^m^−2^ fluorescent light (unless otherwise stated) for three days prior to transfer to 18 °C under dim light. Liquid cultures were inoculated from freshly restreaked plates into 20 mL volumes prior to further subculture for transformation or analysis. Flasks were incubated at 25 °C under 24 h ~50 μE s^−1^m^−2^ fluorescent light (unless otherwise stated) and shaken at 125 rpm. Analytical cultivations were conducted using an Algem HT24 photobioreactor (Algenuity, Stewartby, UK), at a range of light intensities, at 25 °C and 125 rpm. Growth tests were conducted by spotting 5 µL of mid-log cells normalized by OD measured at 750 nm, together with serial dilutions, onto HSM or TAP solid media as appropriate.

### 2.2. Construction of Transformation Plasmids

All plasmids were built using the STEP Golden Gate assembly process (manuscript in preparation), which is based on the StStA method [26]. Briefly, individual expression elements (‘parts’) were cloned as Level 0 plasmids, which were then combined by *Sap*I-mediated Golden Gate assembly into Level 1 expression cassettes. These were then combined with Level 1 flanking region parts for homologous recombination in a further *Bsa*I-mediated Golden Gate reaction to yield the final Level 2 transformation plasmids as depicted in Figure 1. Plasmid details are supplied in Genbank format in the Appendix A.

### 2.3. Transformation of C. reinhardtii

*C. reinhardtii* was transformed by particle bombardment as described previously [27] using a Biolistic PDS-1000/He Particle Delivery System (Bio-Rad, Hercules, CA, USA). Four hundred milliliters of transformation cultures were grown to early log phase (1–2 × 10^6^ cells/mL) and harvested by centrifugation at 3000× *g*. Cells were resuspended in sterile TAP or HSM media (as appropriate) to a density of 2 × 10^8^ cells/mL, and 300 μL was plated directly onto TAP plates containing 100 μg/mL ampicillin. DNA was coated onto 0.4 μm gold DNAdel carrier particles (Seashell Technology LLC, La Jolla, CA, USA) at 5 μg DNA/mg gold, with 0.5 mg of particles used per bombardment, with each transformation carried out in triplicate. Plates were sealed with parafilm and incubated under dim light overnight at 25 °C. For transformations based on *aadA* selection, cells were resuspended in 3 mL TAP per plate, and 1.5 mL were replated onto TAP plates containing either spectinomycin or streptomycin at 200 and 50 µg/mL, respectively. Plates were incubated at 25 °C in either light or dark conditions as appropriate. For transformations based on *psaA-3* restoration, plates were moved from dim light to high light (100 μE s^−1^m^−2^ fluorescent light) following an overnight recovery period. Transformant lines were restreaked to single colonies three to four times on selective media to obtain homoplasmic lines.

### 2.4. Genotyping of Transformant Lines

Lines were assessed by PCR analysis using a three-primer strategy as described previously [14]. In each case, a forward flanking primer was designed outside of the left homology arm used for transformation, with reverse primers designed within the parental and transformed cassettes, respectively. Details of the primers used are given in the Appendix A. Insertion of the target transgene was further confirmed by PCR amplification of the expression cassette followed by Sanger sequencing of the PCR product. Primers used for PCR and sequencing are detailed in Appendix A. 

### 2.5. Luciferase Assays

Expression of *lucCP* in transformant lines was assayed using the Steady-Glo Luciferase Assay System (Promega UK Ltd., Southampton, UK). Fifty-milliliter cultures for the luminescence assay were inoculated from 20 mL starter cultures and grown to mid-log phase (3–5 × 10^6^ cells/mL). Cultures were normalized by OD at 750 nm by diluting them with TAP to that with the lowest absorbance value, and 100 μL was mixed with 100 μL of the sample assay buffer in a 96-well plate format. Following five minutes of incubation at room temperature, luminescence was read using a FLUOstar Omega Microplate Reader (BMG Labtech, Buckinghamshire, UK) over the full visible spectrum (400–700 nm).

## 3. Results

### 3.1. Creation of a PSI Knockout Mutant and Identification of an Adjacent Neutral Locus for Foreign Gene Insertion

To create a ΔPSI mutant, we targeted *psaA*, which encodes one of the two core subunits of photosystem I. In *C. reinhardtii*, *psaA* is split into three separate exons by two trans-spliced introns, with *psaA-3* being the largest exon [28]. Previous studies have shown that point mutations in *psaA-3* or its complete deletion can prevent PSI assembly or function, resulting in a light-sensitive phenotype [29]. This gene was therefore selected for the creation of a new PSI knockout mutant in which the deletion extends beyond the transcribed region of *psaA-3* to an intergenic site that could be used as a neutral locus for subsequent transgene insertion. Although a site downstream of *psaA-3* has previously been identified as a suitable locus for transgene insertion [27], we investigated the upstream intergenic region between *wendyII* and *psaA-3* since this would allow deletion of the 5′ end of the *psaA-3* exon preventing its transcription. This would avoid creating a ΔPSI strain that retains the metabolic burden of synthesizing and then degrading a truncated PsaA subunit.

To confirm that the upstream region could serve as a neutral locus for transgenes, we initially created a transformant line in which the *aadA* marker [30] was inserted at a site 120 bp upstream of the start of *psaA-3* transcription, and in the opposite orientation (Figure 1A). Transformation was achieved by microparticle bombardment using plasmid pHT158 in which *aadA* is linked to the *atpA* promoter/5′UTR and *rbcL* 3′UTR/terminator as previously described [30] and flanked by left and right homology arms to target the marker to the site (Figure 1A).

For several of the pHT158 transformant lines, the correct insertion of the *aadA* cassette into the plastome, and subsequent homoplasmy was confirmed by PCR analysis. The ability to obtain a homoplasmic state (Appendix A), together with the absence of any negative growth phenotype, indicated that insertion at the upstream locus does not disrupt any important plastome elements and is, therefore, a suitable site for transgene insertion. The pHT158 plasmid was thus redesigned with a new left homology arm (plasmid pHT161) such that recombination with the plastome would remove a 600 bp region starting from the insertion site and including the entire *psaA-3* promoter, the intron/exon splicing region, and the first 50 codons (Figure 1B). Transformation was conducted as before with the exception that selection was conducted in complete darkness due to the photosensitive nature of the desired transformants. Numerous antibiotic-resistant colonies were recovered, and four were restreaked under selection in the dark to obtain homoplasmic lines. One representative line was chosen and named HNT6 (Appendix A).

### 3.2. Strain HNT6 Is Incapable of Phototrophic Growth and Is Sensitive to High Light

The photosensitivity of HNT6 compared to the parental line CC-1690 was assessed by measuring growth in acetate-containing medium using a multiplex lab-scale photobioreactor with white light illumination set at 0, 20, and 200 μE m^−2^s^−1^ (Figure 2A). The phototrophic phenotype was determined by growth on solid media with or without acetate, and in darkness or under continuous white light of 100 μE m^−2^s^−1^ (Figure 2B). Both analyses demonstrated the light-sensitive phenotype expected of a ΔPSI mutant, with the inability of HNT6 to grow on minimal media, confirming its photosynthetic deficiency.

### 3.3. HNT6 Can Serve as a Transformation Recipient Using High Light as the Selection

To assess whether HNT6 could function as a next-generation recipient line, a test transformation was conducted. Here, HNT6 was transformed with plasmid pHT191, which contains the WT *psaA-3* sequence together with a transgene cassette cloned into the insertion site upstream of the gene. The cassette comprises a codon-optimized sequence (*lucCP*) encoding the firefly luciferase reporter [31] fused to the *C. reinhardtii rrnS* promoter, *psaA-1* 5′UTR, and *rbcL* 3′UTR elements, as shown in Figure 1C. Transformation with pHT191 was predicted to restore the WT *psaA-3* genotype and introduce a functional lucCP gene into the plastome. Two different selection strategies were tested: The restoration of phototrophy by plating on minimal media (=HSM) and incubation under ~100 μE m^−2^s^−1^ white light, or the restoration of light tolerance by plating on acetate-containing medium (=TAP), also under ~100 μE white light. A negative control experiment involving no transforming DNA was also carried out in order to determine whether colonies would arise on the TAP medium through secondary mutations that suppress the light-sensitive phenotype. 

Colonies were visible after 6 and 12 days for the TAP and HSM platings, respectively, with those on the TAP plates growing faster owing to faster mixotrophic growth on the acetate-containing medium compared to phototrophic growth on the minimal medium. After two weeks, the plates were photographed, and as shown in Figure 3, significantly more colonies (by a factor of 10) were obtained following light selection on TAP compared to phototrophic selection on HSM. Similar colony numbers and the 10-fold difference between the TSP and HSM results were consistently observed in replicates for each condition. For the –DNA control plated on TAP, a few ‘false-positive’ colonies were observed, typically around the edge of the plate where light levels might be lower due to shading and therefore untransformed HNT6 escapes the photoinhibition. Since biolistic transformation typically results in transformant colonies in the center of the plate where bombardment is most effective [32], a few of the colonies at the edges of the TAP+DNA plate likely also represent false positives.

Taken together, the data indicate effective light-mediated inhibition of non-transformed HNT6 cells on mixotrophic medium, with higher transformation efficiency and faster colony recovery than phototrophic selection on minimal medium. The percentage of false positives on the TAP medium is low (<10%) so this is not a major issue as several primary colonies are typically picked for subsequent analysis. Indeed, multiple colonies from both the HSM and TAP transformation plates were restreaked on both TAP and HSM plates and incubated in the light. As expected, all those from the HSM plates grew on HSM, and the majority of those from the TAP plates grew on HSM. Those that failed to grow on HSM also displayed poor growth on TAP when compared to those that grew on HSM (Appendix A). This was also seen for all colonies from the TAP–DNA control plates and therefore provides a simple method for identifying false-positive colonies following light-mediated selection on TAP medium.

### 3.4. Analysis of HNT6 Transformants

Independent colonies from each medium were streaked repeatedly to single colonies on either TAP or HSM plates. After each restreaking, a three-primer PCR analysis of colonies was carried out as illustrated in Figure 4A to determine the state of the polyploid plastome (i.e., heteroplasmic or homoplasmic for the engineering plastome [14]). It was observed that although the cultures on TAP required one or two more restreakings to reach homoplasmy, likely due to the ~two-fold higher plastome ploidy seen in cells grown under mixotrophy compared to phototrophy [33], the much faster mixotrophic growth rate meant the overall time required to achieve stable homoplasmic lines was lower: Homoplasmy was reached in 9–12 days on TAP vs. 10–15 days on HSM. A final PCR analysis of five lines is shown in Figure 4B, confirming their homoplasmy.

One HNT6::Luc line was selected as a representative transformant and the luciferase activity was assayed using as a negative control the original WT strain (CC-1690) used to create HNT6. A significant luminescence signal was observed for HNT6::luc whereas no background luminescence was detected in the WT strain (Figure 4C) demonstrating that lucCP is being expressed in the chloroplast.

The HNT6::LucCP line was also assessed for light sensitivity and photosynthetic ability by comparing growth to the WT strain under different light levels, as shown in Figure 2B and Figure 5 above. Essentially, the transformant showed a wild-type phenotype confirming that photosynthetic function in HNT6 has been restored by transformation with the cloned *psaA-3* gene, and that restoration is not impeded by the insertion of a transgene upstream of *psaA-3*.

### 3.5. HNT6 Can Be Used for Rapid Parallel Transformation of C. reinhardtii Yielding Marker Free Lines

As a further demonstration of the ability to rapidly generate chloroplast transformant lines using HNT6, the *lucCP* cassette in plasmid pHT191 was redesigned such that the promoter element was replaced with that from each of five other endogenous genes: Namely, from the photosynthesis genes *atpA*, *psaA-1*, *psbA*, and *rbcL*, and from the unassigned gene contained within the *wendy* element [34]. These five plasmids were each transformed into HNT6 using restoration of light tolerance on TAP plates as the selection, as described above. This allowed the generation of homoplasmic lines within several weeks. Two independent lines for each promoter were assessed for luminescence yield, together with the original HNT6::luc line and CC-1690 as controls. As shown in Figure 6, different levels of *lucCP* expression were obtained reflecting both the relative strengths of the different chloroplast promoters [35] and also the requirement of additional *cis* elements beyond the core promoter regions for the expression of some endogenous genes such as *psbA* and *rbcL* [36].

## 4. Discussion

The *C. reinhardtii* chloroplast is actively being explored as a GRAS (‘Generally Recognised As Safe’) and sustainable platform for the commercial production of recombinant proteins [4,5,37]. Nevertheless, the presence within the engineered chloroplast of multiple copies of an antibiotic resistance marker is undesirable, both from a regulatory [7] and a metabolic burden perspective [8]. ‘Marker-free’ strategies based on the phenotypic rescue of a mutation within the plastome are preferable, and various photosynthetic mutants with point mutations or deletions in chloroplast genes have been used for the selection of transformants [8]. However, each of these mutants has drawbacks in terms of their suitability and ease of use. In this work, we describe a new recipient strain, HNT6, which addresses these drawbacks. HNT6 was created by the partial deletion of the *psaA-3* gene to generate a cell line lacking in PsaA, and thus unable to assemble Photosystem I. In the absence of this photosynthetic complex, cells display significant photosensitivity, and therefore this mutant phenotype can be exploited by using the wild-type *psaA-3* as a selectable marker for the rescue of light tolerance whilst still maintaining the advantages of much faster growth of *C. reinhardtii* under mixotrophic conditions. Furthermore, we have found that the transformation of HNT6 using restored light tolerance, rather than restored phototrophy, as the selection regime results in significantly higher transformation rates. As with TN72, our previously created recipient strain [14], HNT6 is designed so that restoration of the deleted photosynthetic gene is always accompanied by targeted integration of the transgene, thereby avoiding lengthy screening of multiple transformant colonies. However, HNT6 was created in a strain that has a wild-type cell wall, whereas TN72 possesses the cell wall deficient nuclear mutation, *cw15*. This not only allows HNT6 to be transformed using the biolistic method, rather than the much less efficient ‘vortexing with glass beads’ method, but also results in more robust transformant lines that are less prone to osmotic and shear stresses during downstream processing [23]. As summarized in Figure 7, HNT6 allows for a more efficient and faster pipeline for the generation of marker-free transgenic lines when compared to TN72 and similar strains that employ phototrophic rescue [8]. This is particularly valuable when screening multiple expression elements or transgene variants as part of a synthetic biology-based program of a ‘design-build-test-learn’ cycle [4]. We have illustrated this with a small ‘proof-of-concept’ study that allowed us to quickly evaluate the relative strength of six different promoter elements driving the expression of a luciferase reporter.

HNT6 is available through the Chlamydomonas Resource Center at the University of Minnesota (www.chlamycollection.org, accessed on 1 June 2023) as strain CC-5937.

## Figures and Tables

**Figure 1 microorganisms-11-01967-f001:**
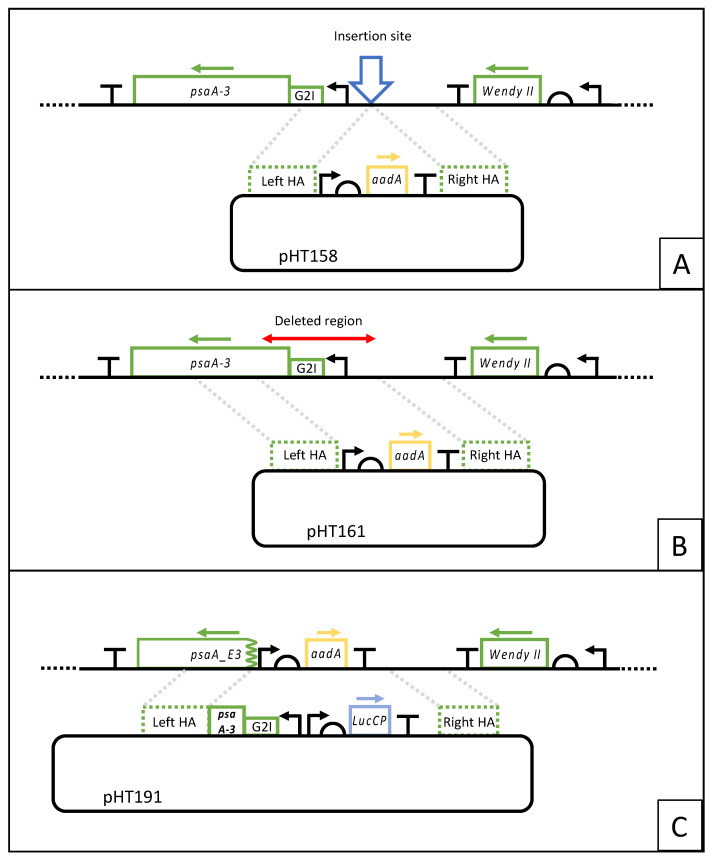
Plasmid constructs and plastome integration strategies for assessment of the chosen insertion site, generation of HNT6, and demonstration of photosynthetic restoration mediated transformation. (**A**) Insertion of the *aadA* cassette in the intergenic region upstream of *psaA-3*. (**B**) Generation of HNT6 by replacement of the *psaA-3* promoter region, the group II intron (G2I), and the first 50 codons of *psaA-3* with the *aadA* cassette. (**C**) Restoration of the wild-type *psaA-3* and co-insertion of the *lucCP* cassette. Green dashed boxes represent the 1.0 kb plastome sequences on the plasmids that serve as homology arms (HA) for recombination with the plastome regions shown in the upper part of each figure.

**Figure 2 microorganisms-11-01967-f002:**
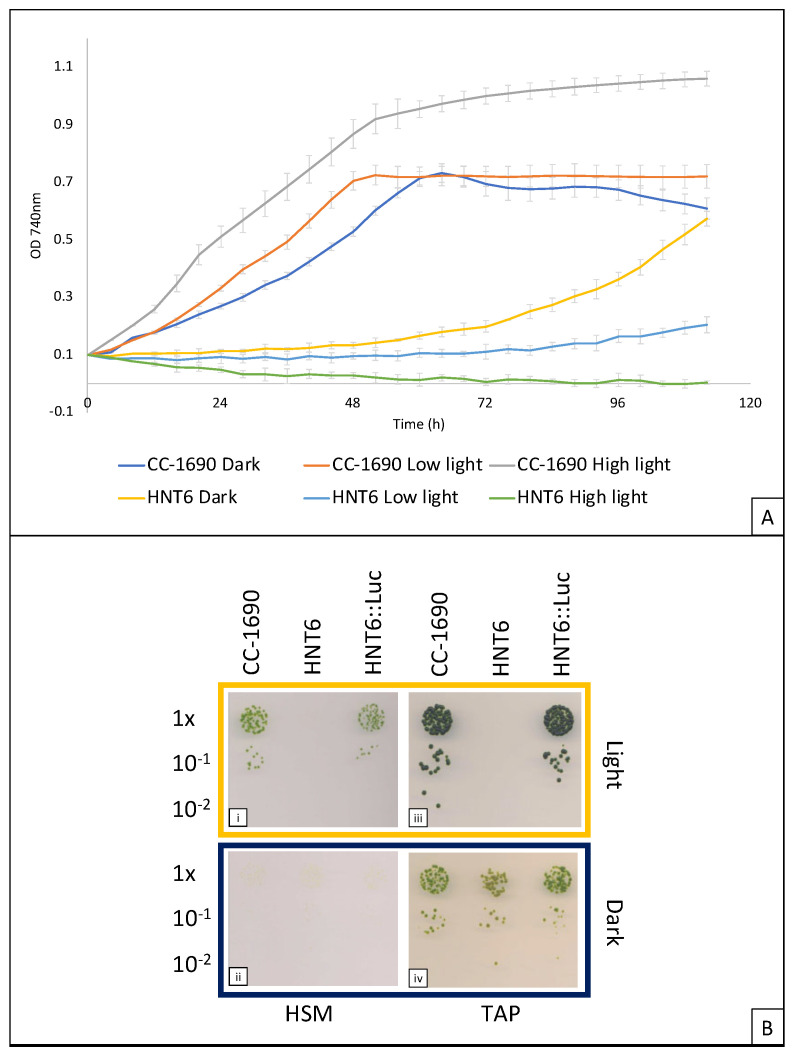
Growth analysis of HNT6 relative to the parental line, CC-1690, and a restored line HNT6::Luc. (**A**) Liquid media trial conducted in an HT24 multiplex photobioreactor. HNT6 is shown to be entirely inhibited by high light (200 μE m^−2^s^−1^), and significantly retarded in low light (20 μE m^−2^s^−1^). In the dark, there is also a marked increase seen in lag phase relative to the parental control, possibly due to poor overall cell fitness. All cultures were run in triplicate on an Algem HT24 photobioreactor and were normalized to an initial OD^740 nm^ of 0.1. (**B**) Spot tests conducted on solid media. HNT6 is shown to be non-photosynthetic by its inability to grow on minimal media in the light in contrast to the WT and the photosynthetically restored line HNT6::Luc (panel i). No photosynthetic growth is seen in dark-incubated plates, as expected (panel ii). HNT6 is also incapable of growing in the light even when provided with acetate as a fixed carbon source, indicating photoinhibition. In contrast, WT and the restored line are unaffected (panel iii). All three lines are capable of growth in the dark in the presence of acetate (panel iv).

**Figure 3 microorganisms-11-01967-f003:**
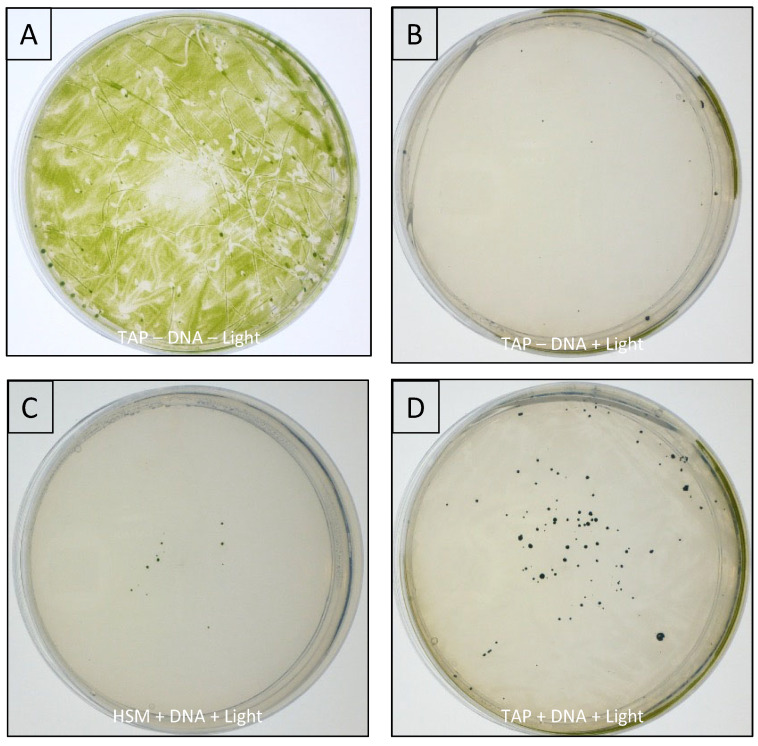
HNT6 transformed with *lucCP* in a photosynthetic restoration vector. (**A**) TAP + DNA + Light (mixotrophic test case): 104 colonies. (**B**) HSM + DNA + Light (phototrophic test case): 13 colonies. (**C**) TAP–DNA + Light (negative control, with selection): 11 colonies. (**D**) TAP–DNA–Light (negative control, no selection): Lawn.

**Figure 4 microorganisms-11-01967-f004:**
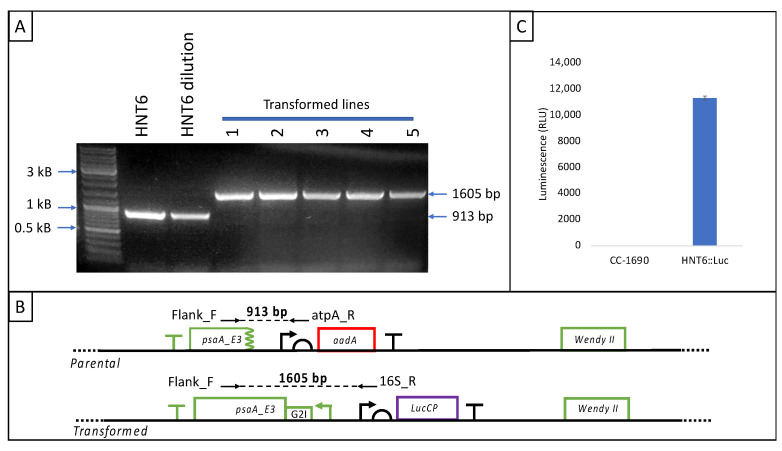
PCR and phenotype analysis of the photosynthetically restored line HNT6::Luc. (**A**) Schematic representation of parental and transformed genotypes showing location of primers used to discriminate between the two genotypes. (**B**) Confirmation of homoplasmy in transformant lines by three-primer PCR. The parental band (913 bp) is seen for HNT6 and remains detectable following a 100-fold dilution of HNT6 genomic DNA (this dilution is intended to approximate less than one plastome copy per chloroplast). In contrast, the transformed band (1605 bp) is seen for all five transformed lines and no parental band is detected. (**C**) Luminescence assay demonstrating firefly luciferase activity in the representative HNT6::Luc strain. The bar shows mean and standard deviation of triplicate readings for this strain.

**Figure 5 microorganisms-11-01967-f005:**
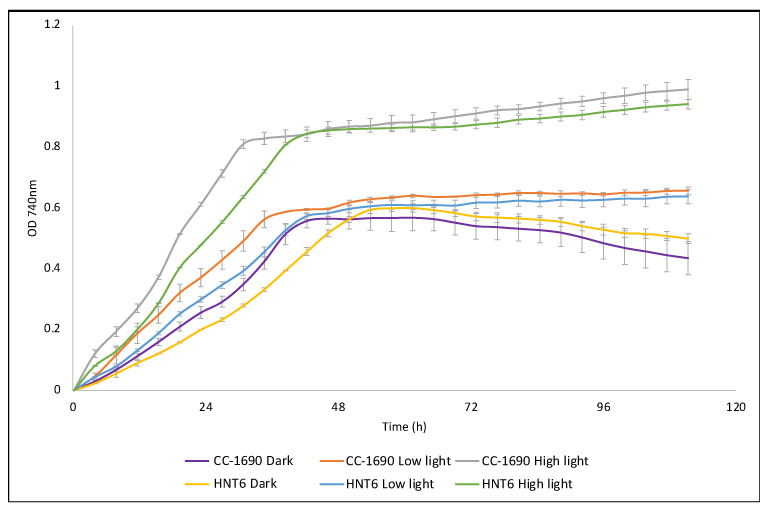
Growth analysis of the HNT6::Luc transformant relative to the original CC-1690 strain. Under high light (200 μE m^−2^s^−1^), low light (20 μE m^−2^s^−1^), and dark conditions, HNT6::Luc displays similar growth to CC-1690 confirming restoration of the WT phenotype. Slight differences are seen during the initial mixotrophic stage, but once the provided acetate is exhausted and the cultures shift to phototrophic growth, any significant variance between the two lines is lost. All cultures were run in triplicate on an Algem HT24 photobioreactor with starting cultures normalized to an OD_740nm_ of 0.1.

**Figure 6 microorganisms-11-01967-f006:**
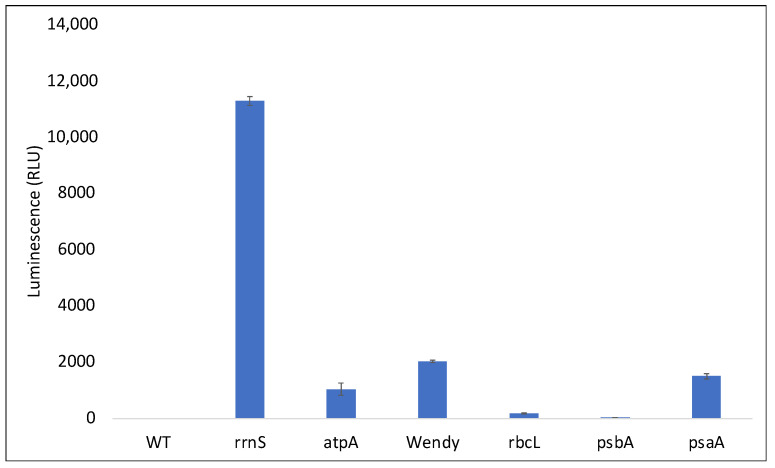
Expression of the *lucCP* reporter gene under six different promoters. In each case, the reporter is fused to the same 5′ and 3′UTR elements (from *psaA-1* and *rbcL*, respectively) with transcription driven by the different core promoter elements. Two representative lines for each of the different HNT6::Luc transformants were grown to late log phase then normalized by OD_750_ and assessed for luciferase activity using a multiplate reader. Bars represent the mean and standard deviation of a single reading for each of the pairs of strains.

**Figure 7 microorganisms-11-01967-f007:**
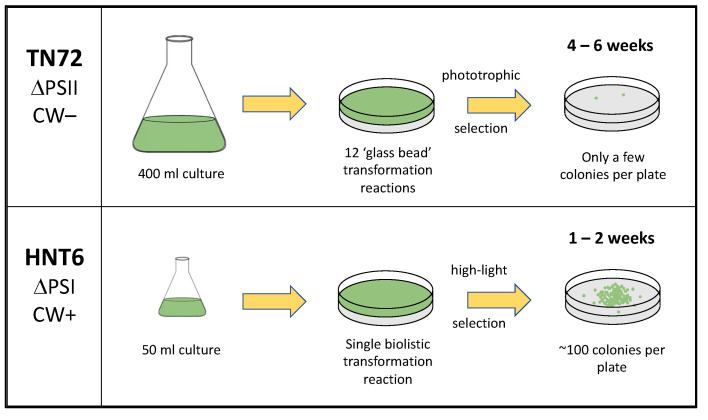
Comparison of TN72 and HNT6 as transformation recipients. With the latter, high-light selection on acetate-containing medium results in visible colonies in 1–2 weeks as opposed to 4–6 weeks for phototrophic selection on minimal medium. Furthermore, both biolistic transformation and selection under high-light result in a significant increase in transformation rates.

## Data Availability

Not applicable.

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
