# Peer review of "Accelerating Chloroplast Engineering: A New System for Rapid Generation of Marker-Free Transplastomic Lines of Chlamydomonas reinhardtii"

_microorganisms, 2023, doi:10.3390/microorganisms11081967_

Round 1

Reviewer 1 Report

This is an interesting manuscript that describes an advance in chloroplast engineering of C .reinhardtii using a new vector/mutant strain combination and a selection regime that will be useful to the research community. The results are convincing and illustrated by appropriate Figures.  The text is well written and contains relevant references.

The authors should consider a number of points:

line 46  replace ‘using in’ with ‘used’

line 88 Wannathong (2016) et al    Insert the date.

line 192   ‘…, it was decided to investigate’.  Consider replacing with,  ‘we investigated’

Fig 1. Indicate the sizes of  the left and right HAs in the legend or Figure. If the figures are to scale consider including scale bars.

Lines 261-271 and Fig. 3.   There can be considerable variation between biolistic bombardments of C. reinhardtii lawns under the same selection regime and using the same strain.  This fluctuation between bombardments can be addressed by replicates. If available indicate the number of replicates (separate bombardments) done for the two regimes: high light TAP vs minimal high light. Include the average number of colonies per plate from multiple bombardments under the two conditions.

Reviewer 2 Report

Taunt et al. describe a novel strategy for producing and screening transformants of the plastome of Chlamydomonas reinhardtii. High level expression of transgenes in C. reinhardtii and other chlorophytes is an important topic with significant industrial and scientific applications. The experimental design is a clever improvement over previously available approaches, and the authors rightly point out that removing the need to use antibiotic resistance for selection reduces the risk of creating antibiotic resistant bacteria by horizontal gene transfer. The manuscript is well written, and should be of interest to many. Lastly, I appreciate that the authors have made their strain available via the Chlamy Center, and they do a good job of providing sufficient details on their approach for other groups to replicate and to apply a similar strategy in other organisms.

My biggest concern regards the claim that the authors are able to achieve homoplastomy. The authors attempt to demonstrate homoplastomy with gels (Fig. 4B) showing the presence/absence of two amplicons (with and without the transgene) produced using a three-primer PCR assay. I do not find this approach to be completely convincing because: (A) the two amplicons may not amplify equally efficiently, especially in the same reaction, which could erroneously imply that the no transgene plastome is not present, and (B) careful manipulation of the number of PCR cycles could artificially give the impression that one amplicon is present and the other is not, and the number of PCR cycles isn't specified anywhere I could find.  

The degree to which they have achieved homoplastomy could be made much more convincingly with qPCR, especially relative to dilution series of known quantities of each amplicon. Such an assay could establish if the primer pairs are working at near 100% efficiency, and are thus comparable. With this analysis, the authors could give a more definitive and quantitative estimate of the ratio of plastomes +transgene versus -transgene over a range of many logs.

I bring this up in part because the plastome without a transgene retains the aadA gene, which could still be a risk for horizontal gene transfer. Additionally, Fig. 3 shows that there are a non-trivial number of false positive colonies in the -DNA control that have presumably escaped photoinhibition. It seems that the most likely explanation for this is that the HNT6 strain is not completely homoplastomic (i.e. retains some plastome copies with an intact psaA-3 gene).

One other major point: In Fig. 6, it seems very odd that the authors see relatively high transgene expression from the Wendy promoter, and very low expression from rbcL and psbA promoters. Was this reproducible? I.e. Do you see consistent levels of Luc activity with multiple transformants using the same promoter? I would be more inclined to attribute differences in transgene expression to inter-transformant variability than the different promoters unless the authors had looked at multiple transformants. It was not clear to me that they had.

Fig. S2 - S4 were missing from the file I received and could not be evaluated.

Minor points:

The luciferase assay in Fig. 4C clearly shows there is more luciferase activity with the transgene than without, but this point would be much better with more context. Can you at least roughly estimate how much transgene is being produced, say in terms or mass or moles of transgene product per cell or per mass of cell pellet?

Fig. 2A and Fig. 3: The labels are too small and too light. Larger font and black text would improve readability.

Fig. 5: the colors of the blue lines for CC-1690 Dark and HNT6::luc Low Light are too similar

Fig. 6: RFU? Do you mean RLU?

Fig. 6 legend: "reporter" versus "report"

Line 46 "used" vs. "using"

Line 69 "...not tailor-made for chloroplast genetic engineering" What does this mean exactly?

Round 2

Reviewer 2 Report

The authors have mostly either addressed my concerns or made a reasonable argument for why they did not.

My remaining concern, that could easily be addressed, and really should be before acceptance is regarding the reproducibility of the luciferase assays in Fig. 4C and Fig. 6. The text indicates that two separate transformants were produced with each construct, and that there were triplicate luciferase assays performed. But this left me with some easily addressed questions about what is being presented. What do the size of the bars indicate in each figure? The mean RLU for three measurements of one transformant (i.e. N=3)? The mean RLU for three measurements of both transformants combined (i.e. N=6)? If just one transformant, what about the other one? What are the error bars? Standard deviation? All of this should be indicated in either the legend or the Methods, but I couldn't find it. Really, the authors should show the mean of three measurements of one transformant side-by-side with the mean from the other transformant.

Author Response

We thank the review for their comments. We have reviewed the original data for the luciferase assays and have clarified this in the manuscript. For figure 4C, the bars represent the mean and SD of triplicate measurements from a single line. For figure 6, the values are the mean and SD from two independent lines, each measured once. We have amended the figure legends to clarify this.